# A Business Strategy Framework for Patent Pooling in the Solar Cell Supply Chain

**Shuo-Min Lee [1], Jen-Shyang Chen [1], Chien-Lin Hsu [2] and Chi-Cheng Chang [3,***

[1]    Department of Business Administration, Ming Chuan University, Taipei 111, Taiwan;
miin9133@yahoo.com.tw (S.-M.L.); jschen@mail.mcu.edu.tw (J.-S.C.)
[2]    Tainan City Council, Tainan City 708008, Taiwan; relation20082012@gmail.com
[3]    Department of Information Management, Lunghwa University of Science and Technology,
Taoyuan City 333326, Taiwan
*    Correspondence: chig@mail.lhu.edu.tw

**Abstract:** In the green energy sector, which includes the solar cell industry, effectively combining the patents of various manufacturers within the supply chain to collectively create sustainable patent market value is of paramount importance. This study aimed to establish a business strategy framework for project-based licensing through the analysis of solar cell patents and the integration of supply chain manufacturers' patents, with the goal of collectively creating market value for patents in a sustainable manner. The research methodology included analysis of the literature and focus group discussions. The study began with an analysis of solar cell industry patents from the past decade using the WIPS Global Patent database to identify projects with the highest patent count. Subsequently, through the use of focus group discussions, the feasibility of a patent pool strategy within the supply chain was explored. Finally, a strategy framework was proposed for project-based patent pool licensing, focusing on the top 15 projects with the highest count of 5579 patents. The results indicate that adopting this strategy framework for supply chain manufacturer patents can result in higher commercial patent value.

**Keywords:** solar cell; patent analysis; business strategy; patent pool; supply chain

## 1. Introduction

The solar cell industry, which falls within the realm of renewable energy, is also a globally competitive "green energy" sector. Patents serve as a pivotal cornerstone in driving the advancement of this industry [1,2]. Within the context of business operations, intellectual property rights play a crucial role in fostering entrepreneurship and act as a vital mechanism for preserving corporate R&D achievements while facilitating sustainable corporate growth [3]. Patents, from a defensive standpoint, protect creators' intellectual property rights from potential infringement. On the market front, the exclusive patent rights held by each manufacturer throughout the supply chain establish barriers to entry, reducing competitive pressures. However, while the active pursuit of "defensive patents", the utilization of patents to create robust barriers against market entry, and the formation of complex "patent thickets" for countering rivals and seeking compensation through infringement litigation, can be viable strategies, they can also hinder industrial progress and societal advancement [4]. Regarding innovation, patent licensing or the pursuit of complementary patents offers patent holders avenues for financial gains [5]. Taiwan's photovoltaic industry boasts well-established upstream, midstream, and downstream industrial chains. In this context, the aggregation of patents from diverse manufacturers within the supply chain is particularly crucial in fostering collective market value derived from patents.

The establishment of a patent pool primarily addresses the challenge posed by blocking patents [6]. According to Bourgeron and Geige [7], patents are important assets of

enterprises, and blocking patents are inevitable. Blocking patents emerge due to the continuous generation of innovations that enhance the original patented technology during research and development (R&D) and innovation processes. Without new patents to cover these enhancements, the progression of innovation could be hindered, leading to these new patents being relegated to a secondary status known as subservient patents [6]. The concept of pre-emptive patenting involves filing a patent application with the primary intention of obstructing the grant of other patents, resulting in the creation of blocking patents [8]. Upon its commercialization, a new patent often finds itself infringing on the original patent, leading to a situation where the original patent effectively obstructs the implementation of the subservient patent. As a result, effectively deploying the new patent in the market becomes challenging [3]. This phenomenon of patent safeguarding is considered an inevitable and common aspect of business operations.

Various approaches to reviewing previous research can be identified as follows:

The legal protection concept: Ho [4] delves into the concept of horizontal competition among members of patent pooling, potentially leading to anticompetitive effects. His work primarily investigates legal disputes arising from patent pooling. Guellec, Martinez, and Zuniga [8] introduce a methodology to identify pre-emptive patent applications, focusing on the practice of pre-emptive patenting. This involves strategically designing patents to create blocking patents that hinder others from filing patents.

The patent layout concept: Within the context of patent layout, Crescenzi, Dyèvre, and Neffke [9] emphasize that engagement of foreign technology leaders in technical poolings and collaborations with less developed local companies could hinder the generation of new patents. Di et al. [10] employ network analysis to reveal a distinct "core-periphery" pattern in the global innovation network. Their findings present a quadrilateral structure with vertices represented by the "US, Japan, Europe, and China mainland", providing a visualized map of the global patent layout and its evolutionary trajectory.

The patent portfolio management concept: Conegundes De Jesus, and Salerno [11] elucidate the evolution and trends in patent portfolio management, presenting a relevant conceptual framework. Hoskisson and Yiu [5], along with Sun and Xu [12], delve into research on complementary patents, and suggest that integrating such patents can enhance technological advancement within an enterprise's research and development (R&D) process. Applying complementary technology to allied enterprises can further amplify these benefits. Sun and Xu's [12] methodology calculates the average complementarity between a given patent and all patents within the same company, facilitating the identification of desired complementary enterprises. Granstrand [13] underscores the necessity of employing diverse complementary intellectual property rights for distinct innovative components, suggesting six patent strategy models that hold instructive value for this study.

Drawing on an in-depth analysis of the relevant literature, the central theme revolves around the legal protection of patents, which is reinforced by patent deployment and the synergy of patent complementarity. The excessive emphasis on the market value derived from a company's additional intellectual property through patents has led to an augmented impact known as "within-effects". Furthermore, competitors holding more intellectual property leads to beneficial "between-effects" [14]. Paradoxically, this phenomenon is less focused on the positive aspects of patent application. A research gap is identified in the limited exploration of methodologies for aggregating patents from supply chain manufacturers to collectively cultivate patent market value. This gap is particularly pronounced within small and medium-sized enterprises grappling with resource constraints in their enterprise supply chains. For these entities, unlocking their patent portfolios could potentially yield significant benefits, enhancing patent utilization and bolstering operational performance. Hence, the core of this research centers on formulating a comprehensive business strategy framework that spans a range of patent projects. The aim is to enable each enterprise along the supply chain to leverage its patent reserves and thus generate patent market value.

The primary focus of this research is to devise a business strategy framework that facilitates project-based licensing within patent consortia. The overarching goal is to

collaboratively and sustainably enhance the market value of patents. To achieve this, the study analyzes solar cell patents, coupled with the integration of patents held by supply chain manufacturers. The research employs the WIPS Global Patent database as its dataset, analyzing both abstracts and complete patent texts from the solar cell industry to identify projects with the highest patent density. Subsequently, through focus group discussions, the feasibility of implementing a business strategy involving supply chain enterprise patent pooling is explored. Finally, the research recommends adopting a business strategy framework for project-based patent consortium authorization, tailored particularly toward projects exhibiting the highest patent frequency. It is important to note that the solar cell manufacturing industry within the scope of this study encompasses manufacturers of solar silicon wafers, batteries, and modules [15]. The proposed business strategy framework takes the form of a conceptual construct with defined business objectives, which can be easily translated into market-driven profits.

## 2. Literature Review

### 2.1. Expectancy Theory

Applying the expectancy theory to the domain of patents involves evaluating their inherent anticipated value. Hsueh and Jheng [16] categorize this value into two distinct dimensions: patent litigation value and patent commercialization value. The patent litigation value reflects the expectation that patents hold value by enabling legal actions against potential infringers and deterring patent infringements. On the other hand, patent commercialization value signifies the potential for patents to be commercialized and applied in practical contexts [16]. Realizing these dual aspects of patent value requires thorough analysis, strategic arrangement, and skillful deployment of patents. Considering market demand, a patent licensing resource that reduces R&D costs and enhances research efforts is viewed as a catalyst for innovation, which is particularly sought after by small and medium-sized enterprises, startups, and innovators.

Aligned with the expectations linked to patent value, the development of a patent strategy underscores the importance of strategic decision-making. Robinson, Jr. and Pearce [17] state that strategic decision-making encompasses six dimensions: strategic matters necessitate top-management decisions, entail significant allocation of firm resources, have a lasting impact on the firm's future prosperity, have a forward-looking orientation, often have multifunctional or multi-business implications, and require an assessment of the firm's external environment. Thus, a strategic management framework is essential to the company's goals of extracting commercial value from patents.

### 2.2. Patent Pool

The patent pool model serves as a strategic approach within patent operations, which is designed to efficiently secure authorization and reduce manufacturing costs in response to the complex nature of industrial technology. Often, the production of a single product requires the incorporation of various existing patents [6]. Patent pooling can be succinctly described as an arrangement among parties that hold intellectual property from different projects, either directly or through the authorization of a third party [18]. A critical implication of patent pooling is that patent holders collectively relinquish their exclusive rights associated with patents to consolidate their patents under a single entity [19]. This concept is further emphasized by Andewelt [19], who suggests that patent pooling involves the aggregation of patents.

Regarding aggregation, patent holders can assign or authorize their patent rights to an independent legal entity or unite as members of a patent pool. Subsequently, the independent legal entity or the pool members collectively authorize the accumulated patent portfolio to authorized parties as a unified package. When a patent pool consists of complementary patents, the entities within the pool forgo competitive rivalry, thus avoiding the risk of participating in joint anti-competitive behavior [6,18].

The Antitrust Guidelines for the Licensing of Intellectual Property [18], jointly published by the Department of Justice and the Federal Trade Commission in 1995, elaborate on circumstances within patent pooling that promote competition. These circumstances include:

- Integrating complementary patents;
- Resolving patent blocking situations (clearly defined blocking positions);
- Reducing transaction costs;
- Mitigating the need for expensive patent infringement litigation;
- Facilitating technology utilization.

In a subsequent update in 2017, the DOJ and FTC's antitrust guidance on IP licensing further emphasized that IP licensing allows firms to combine complementary factors of production, often leading to pro-competitive outcomes. As stated, "the Agencies recognize that intellectual property licensing allows firms to combine complementary factors of production and is generally competitive" [18]. As such, patent pools have garnered increasing attention in recent times.

### 2.3. Strategy of Project-Based Authorization

Within the realm of strategic patent deployment, a significant approach is encapsulated in strategic patents, representing a pivotal avenue for strategic utilization [13]. iKnow [20] underscores the crucial need for enterprises (or conglomerates) to conduct a comprehensive patent audit when implementing strategic growth at a global scale. This audit mitigates the considerable risk of potential infringement lawsuits. The essence of a patent audit lies in meticulously organizing, scrutinizing, categorizing, and mapping patents through a systematic process. This allows enterprises to systematize and enhance their original patent portfolio, effectively establishing an intelligence resource planning (IRP) system, which, in turn, enables them to unlock the maximum value of their patents through strategic implementation and exchange in the commercial market. iKnow [20] introduces a patent value inventory represented by a radar chart, as depicted in Figure 1 below, offering a vivid interpretation of this process. The convergence of numerous patents with the foundational elements of the commercial market underscores the strategic importance of forming patent pools as a key approach to patent utilization for enterprises.

The concept of project-based licensing draws inspiration from the domain of complementary patents, as outlined by Sun and Xu [12]. This involves harnessing a patent pooling database formed through complementary patents, followed by tailoring project-specific authorization to enterprises or individuals seeking to incorporate authorized patents within their product or service research and development endeavors. Patents possess four key performance attributes: energy, hardware cost, time, and space [1]. Therefore, when devising strategies for project-based licensing, the allocation of licenses over time, commonly referred to as a year-by-year licensing method, can be a pertinent consideration for the "time" attribute.

When contemplating the adoption of the patent joint venture model to meet the needs of small and medium-sized enterprises, an essential consideration lies in the expansion of demand and in opening avenues on the supply side, accordingly. Incentivizing small and medium-sized enterprises and innovators to invest in pioneering R&D, mitigate R&D costs, and overcome barriers to accessing pre-existing patents in the research and development continuum can be effectively achieved through the utilization of R&D project-based licensing methods. This strategy involves a patent licensing approach that centers on individual projects as the unit of calculation. It is tailored to the requisites of enterprises or individual innovators embarking on novel product or service research and development initiatives.

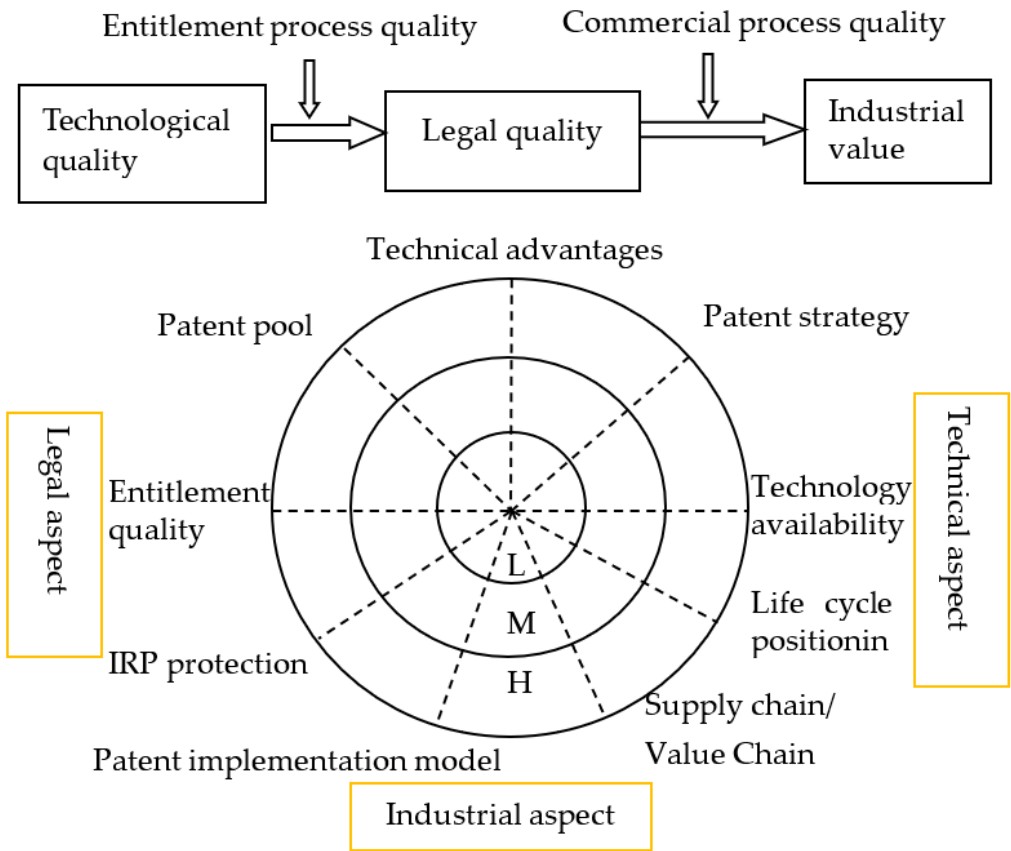

**Figure 1.** Radar chart of patent value inventory. Source: adapted from iKnow [20].

### 3. Methodology

*3.1. Literature Analysis*

This phase involved an exploration of patent pooling within the solar cell industry, along with the relevant literature. Additionally, a preliminary business strategy framework for patent-pooling operations was formulated to serve as the foundation for subsequent focus group discussions.

*3.2. Focus Group Methodology*

The focus group method encompassed two distinct meetings. The solar photovoltaic industry primarily comprises the "solar cell manufacturing industry", which also encompasses "other electric equipment and supplies manufacturing", "engineering services and related technical consulting", and "electric power supply" [14]. This industry is categorized into upstream, midstream, and downstream segments. The upstream segment includes silicon materials and wafers from a single company, while the midstream involves 17 companies manufacturing solar cells and 18 companies producing solar photovoltaic modules. The downstream segment consists of 58 companies producing inverters and components, along with 245 companies focusing on system engineering. Recognizing the interconnectedness of the solar cell manufacturing and photovoltaic industries, a total of 14 managerial-level representatives from these segments were invited to partake in discussions (outlined in Table 1). The focal point of these discussions was the business strategy framework proposed in this research, with a specific focus on assessing its feasibility. The procedural approach adhered to the methodology outlined by Bates [21]. During the implementation process, there is a focus on the research team's initiative to initially present the research backdrop, delve into patent analysis, and outline the research design along with preliminary analytical findings. Moreover, there is thoughtful discussion about the feasibility of the strategic framework from a business perspective.

**Table 1.** Distribution of experts in the solar cell industry focus group.

| Manufacturers' Positions in the Supply Chain | Chief Products | Number of Companies | Number of Companies Interviewed (%) |
|---|---|---|---|
| Upper-stream firms | Silica materials, silicon wafer materials | 1 | 1 (7.14%) |
| Midstream firms | Solar cells<br>Solar cell modules<br>Thin-film solar cell modules<br>Dye-sensitized solar cells<br>Concentrator solar cell modules | 35 | 7 (50%) |
| Lower-stream firms | Solar photovoltaic system<br>Solar photovoltaic converters<br>Sales channels/suppliers of solar photovoltaic products | 303 | 6 (42.86%) |
| Total | | 339 | 14 (100%) |

Source: IDBMEA [15].

*3.3. Patent Analysis*

The primary objective of analyzing patent abstracts and full texts within the solar energy industry is to establish the groundwork for the development of a business strategy framework, building upon the findings from the literature analysis phase. During patent analysis, the process adheres to the structure of the International Patent Classification (IPC). The IPC organizes patents into a hierarchical framework consisting of five levels: section, class, subclass, main group, and group. The patent documents are separated into eight distinct categories: A (human necessities), B (performing operations; transporting), C (chemistry, metallurgy), D (textiles; paper), E (fixed constructions), F (mechanical engineering; lighting; heating; weapons; blasting), G (physics), and H (electricity) [22].

Hence, when selecting the target database and the chosen analysis indicators, the WIPS GLOBAL SmartCloud database was chosen as the retrieval platform for this study. It employed comparative analysis, classification code analysis, and activity index analysis. The retrieval approach involved gathering full-text patents that fall within the vertical industries of the solar cell sector: upstream, midstream, and downstream. The retrieval timeframe spanned a decade, from 23 February 2013 to 23 February 2023. Full-text patents were acquired for subsequent analysis. A detailed investigation of patents related to solar cells within the midstream solar energy industry was performed.

**4. Results and Discussion**

*4.1. Results*

4.1.1. Overall Patent Landscape in the Solar Cell Industry

Using the product classification names corresponding to the vertical segments of the solar cell industry—upstream, midstream, and downstream—as search criteria, the patent counts for each specific product or service were extracted and consolidated in Table 2. These counts were employed as retrieved keywords, and their primary sources were obtained from IDBMEA [15]. The key product categories included silicon materials, silicon wafer materials, solar cell, solar cell module, thin-film solar cell module, dye-sensitized solar cell, concentrating solar cell module, solar PV system, solar PV power, and solar PV power converter. The analysis results indicated that the solar PV system, silicon materials, silicon wafer materials, and solar PV power converter product series possessed the highest number of patents.

**Table 2.** Number of patents in the solar energy industry in the past 10 years.

| Retrieved Keywords: Silicon Materials, Silicon Wafer Materials, Solar Cell, Solar Cell Module, Thin-Film Solar Cell Module, Dye-Sensitized Solar Cell, Concentrating Solar Cell Module, Solar PV System, Solar PV Power, Converter, Solar PV Access/Supplier | | | |
|---|---|---|---|
| | | | **Number of Patents** |
| | Upstream | Silicon materials | 31,212,755 |
| | | Silicon wafer materials | 31,382,887 |
| | Midstream | Solar cell | 10,438,088 |
| | | Solar cell module | 20,007,918 |
| | | Thin-film solar cell module | 27,580,647 |
| | | Dye-sensitized solar cell | 10,438,359 |
| | | Concentrating solar cell module | 25,084,359 |
| | Downstream | Solar PV system | 31,508,226 |
| | | Solar PV power converter | 31,349,698 |
| | | Solar PV access/supplier | 2,075,363 |
| Retrieval Period | 23 February 2013~23 February 2023 | | |
| Retrieval Database | WIPS Global Database | | |
| Retrieval Country | US, EP, PCT, CN, JP, KR, CA, AU, DE, GB, FR, IT, TW | | |

Source: [23,24].

### 4.1.2. Analysis of Solar Cell Patents

A dataset comprising solar cell patents from the past 10 years was compiled, resulting in a total of 221,078,300 patents (see Table 1). A close evaluation of the abstracts of these solar cell patents revealed that the two categories with the most substantial global patent counts are "Wireless Communication" and "Base Station". The comprehensive distribution of patents is presented in Figure 2.

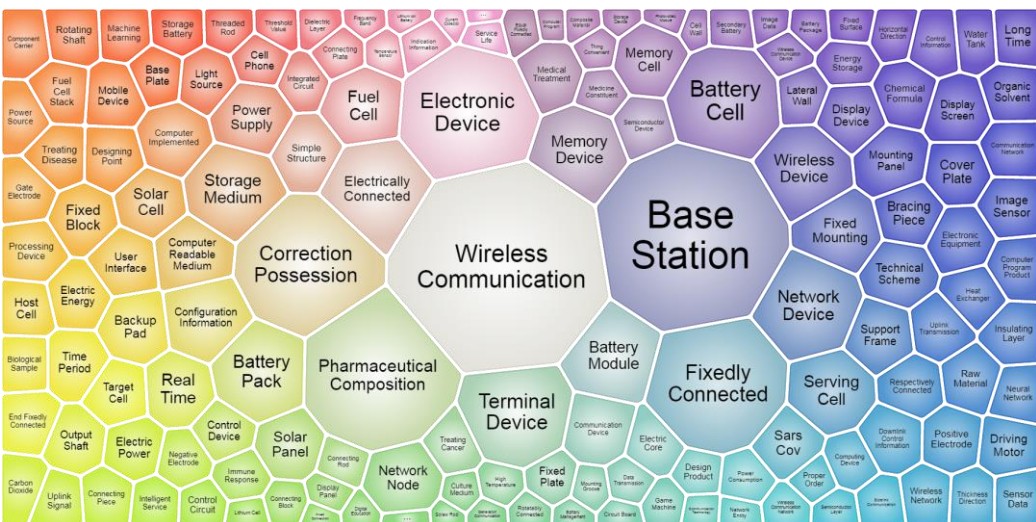

**Figure 2.** Overall patent distribution of solar cells from 23 February 2013 to 23 February 2023. Source: Analysis from [23,24].

A thorough examination of the intricate solar cell patents revealed that the fifteen most prominent solar cell patent projects, comprising a considerable total of 5579 patents, encompass a comprehensive tally of 3297 patent documents. This detailed information is presented in Table 3 below.

**Table 3.** Ranking of patents in the solar cell field.

| No. | Keyword | Frequency | No. of Documents |
|---|---|---|---|
| 1 | Wireless Communication | 669 | 591 |
| 2 | Base Station | 935 | 461 |
| 3 | Pharmaceutical Composition | 373 | 314 |
| 4 | Electronic Device | 428 | 255 |
| 5 | Fixedly Connected | 417 | 249 |
| 6 | Correction Possession | 239 | 239 |
| 7 | Battery Cell | 480 | 180 |
| 8 | Terminal Device | 426 | 166 |
| 9 | Electrically Connected | 217 | 142 |
| 10 | Network Device | 244 | 134 |
| 11 | Battery Pack | 254 | 124 |
| 12 | Memory Device | 217 | 113 |
| 13 | Battery Module | 289 | 112 |
| 14 | Storage Medium | 112 | 111 |
| 15 | Wireless Device | 279 | 106 |
| | Total | 5579 | 3297 |

Source: Analysis from [23,24].

A.    Analysis of Classification Codes.

Further analysis of solar cell patents through the lens of classification codes presents a comprehensive picture of the quantity and proportional distribution of patents within each classification. The breakdown is delineated as follows:

- A (human necessities): 17,485 (18%);
- B (performing operations, transporting): 10,746 (11%);
- C (chemistry, metallurgy): 15,961 (16%);
- D (textiles, paper): 436 (0%);
- E (fixed constructions): 1954 (2%);
- F (mechanical engineering, lighting, heating, weapons, blasting): 3736 (4%);
- G (physics): 19,922 (20%);
- H (electricity): 29,088 (29%).

The patent counts for the leading 15 countries or regions boasting the highest number of solar cell patents, as well as the respective percentages out of the total global patent count of 221,078,300, are outlined in Table 4, with particular emphasis on the top fifteen countries worldwide, in light of their substantial patent holdings. This emphasis aims to underscore the potential contributions of these countries or regions in advancing viable commercial models and, consequently, fostering significant growth in their industrial sectors.

B.    Analysis of Activity Index (by Country).

The Activity Index (AI) pertains to the proportional concentration of patents held by applicants (countries) within specific technological domains. The AI is determined by dividing the number of specific applicants (countries) in a technological field by the total number of patents within that same field. In essence, the AI can be expressed as the total number of patents held by a particular applicant (country) divided by the overall total number of patents. The formula is shown in Scheme 1. Hence, the AI signifies a relative ratio and does not directly indicate patent quantity.

AI values greater than one denote a higher relative concentration of patents owned by the applicant (country) in a specific technological realm. Conversely, AI values less than one are indicative of a lower relative concentration. As exemplified in Figure 3, using the United States as a case study, technological categories such as A61 (human necessities), H04 (electricity), and G06 (physics) exhibit AI values greater than one. Japan, on the other hand, demonstrates a significant AI concentration in the A61 category.

**Table 4.** Top 15 countries/regions for solar cell patents.

| No. | Country/Regions | Country Code | Number of Patents | Percentage of Global Patents (%) |
|-----|-----------------|--------------|-------------------|----------------------------------|
| 1 | United States | US | 172,319 | 0.078 |
| 2 | Mainland China | CN | 93,948 | 0.042 |
| 3 | PCT | WO | 62,936 | 0.03 |
| 4 | EP | EP | 52,884 | 0.024 |
| 5 | Japan | JP | 46,331 | 0.021 |
| 6 | Korea | KR | 39,920 | 0.020 |
| 7 | Australia | AU | 22,350 | 0.010 |
| 8 | Canada | CA | 19,876 | 0.009 |
| 9 | India | IN | 15,448 | 0.007 |
| 10 | Taiwan | TW | 12,608 | 0.006 |
| 11 | Brazil | BR | 10,219 | 0.005 |
| 12 | Israel | IL | 7079 | 0.003 |
| 13 | Germany | DE | 6338 | 0.0029 |
| 14 | Mexico | MX | 4502 | 0.002 |
| 15 | Singapore | SG | 4163 | 0.0019 |
| Total and percentage | | | 570,921 | 0.258 |

Source: Analysis from [23,24].

$$AI = \frac{\text{Number of specific applicants (countries) in a specific technology field / total number of patents in a specific technology field}}{\text{Total number of patents of the specific applicant (country) / Total number of patents}}$$

**Scheme 1.** Calculation Formula of the Patent Activity Index (AI). Source: [24].

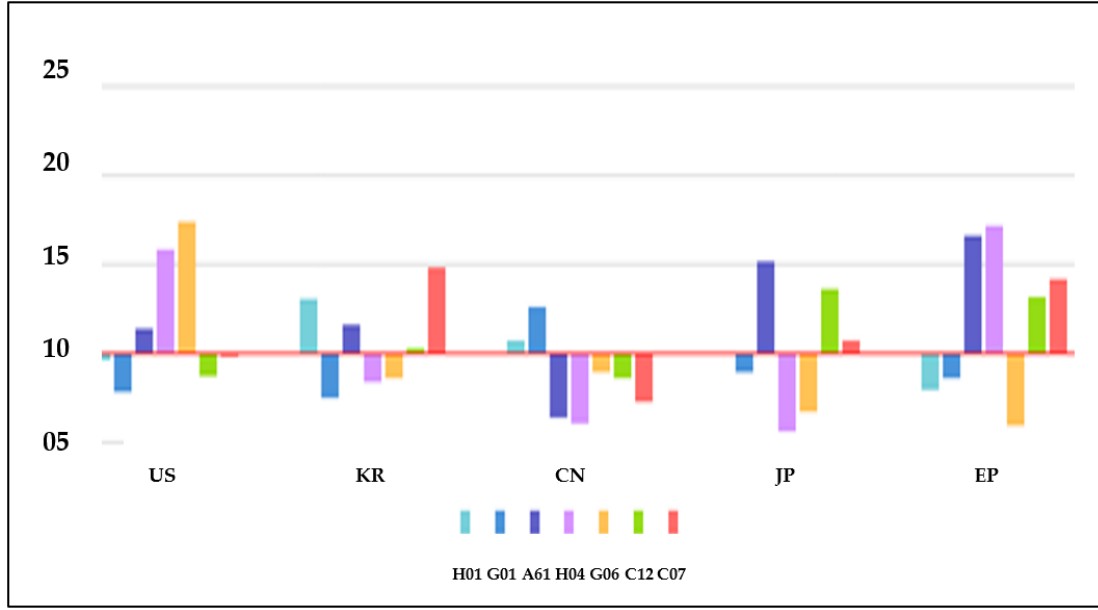

**Figure 3.** Concentration of solar cell patents in specific fields by major countries. Source: Analysis from [23,24].

Please refer to Figure 3 for a visual representation of AI values greater than one in specific technological categories across different countries. The explanation accompanying the figure offers insights into the calculation and interpretation of AI values.

Please find the distribution of Activity Index values greater than one for solar cell patents in specific fields among major countries presented in Tables 5 and 6 below. This table showcases the patent strengths of various countries within the solar cell industry, highlighting their expertise in specific fields as outlined:

**Table 5.** Distribution of activity index values > 1 for solar cell patents in specific fields among major countries.

| Major Countries/Regions | Notable Categories with AI > 1 |
| --- | --- |
| United States | A61 (medical or veterinary science, hygiene), H04 (electric communication technique), G06 (computing; counting) |
| Japan | A61 (medical or veterinary science, hygiene), C12 (chemistry; metallurgy), C07 (organic chemistry) |
| South Korea | H01 (basic electric elements), A61 (medical or veterinary science, hygiene), C12 (chemistry; metallurgy), C07 (organic chemistry) |
| Mainland China | H01 (basic electric elements), G01 (measuring; testing) |
| European Union | A61 (medical or veterinary science, hygiene), H04 (electric communication technique), C12 (chemistry; metallurgy), C07 (organic chemistry) |

Source: Analysis from [23,24].

**Table 6.** Activity index of major countries in specific fields for solar cell patents.

| Country/Regions | H01 | G01 | A61 | H04 | G06 | C12 | C07 |
| --- | --- | --- | --- | --- | --- | --- | --- |
| US | 0.95 | 0.78 | 1.13 | 1.58 | 1.73 | 0.87 | 0.98 |
| KR | 1.3 | 0.74 | 1.15 | 0.83 | 0.85 | 1.02 | 1.47 |
| CN | 1.06 | 1.25 | 0.63 | 0.6 | 0.89 | 0.86 | 0.72 |
| JP | 0.99 | 0.89 | 1.51 | 0.56 | 0.67 | 1.35 | 1.07 |
| EP | 0.79 | 0.85 | 1.65 | 1.71 | 0.59 | 1.31 | 1.41 |

Source: Analysis from [23,24].

Refer to Table 5 for a comprehensive breakdown of Activity Index values greater than one for solar cell patents in specific fields among major countries.

*4.2. Discussion*

4.2.1. Feasibility of Establishing a Patent-Pooling Information Platform

The outcomes of the patent analysis underscore the significance of focusing on the top fifteen solar cell patent projects, which exhibit substantial patent volume and demonstrate the research and development demand within the supply chain. Consequently, opting to target these patents, which are characterized by considerable market economic scale, for the patent pooling information platform increases the likelihood of achieving successful outcomes. Classification code analysis indicates that patents are predominantly distributed across three categories: H (electricity) with 29,088 (29%), G (physics) with 19,922 (20%), and A (human necessities) with 17,485 (18%). Similarly, the activity index (country) analysis underscores distinct patent advantages among different countries within the solar cell industry. Thus, the establishment of specialized patent-pooling information platforms for each country or region, tailored to their unique supply chain requirements, emerges as a strategic proposition. By adhering to the principle of territoriality in a patent application and scope, patent rights holders can choose to participate in local or national patent pools as high-value contributors. This approach enhances the contribution of patents to industries and society, amplifies individual patent capital gains through authorization, and activates intellectual creations, consistent with Conegundes De Jesus, and Salerno's [11] perspectives on patent portfolio management frameworks.

### 4.2.2. Enhancing Patent Commercialization Value through Project-Based Authorization

The commercial framework of the patent-pooling information platform, underpinned by a project-based authorization approach, rests on the collective generation of market value. This approach, in alignment with Bridoux, Coeurderoy, and Durand's [25] concept of "cooperative and nonhierarchical collaboration", holds the potential for comprehensive benefits. Insights from focus groups further emphasize that small and medium-sized enterprises, despite harboring strong innovation and research targets, often grapple with financial constraints and resource inadequacy. Additionally, the hindrance imposed by pre-existing restrictive patents curbs their commitment to research and development, impeding innovation outputs and inhibiting societal and industrial advancement. In response, a project-based authorization model for research and development can be tailored to meet market needs. This perspective echoes Hsueh and Jheng's [16] discourse on the commercialization and application value of patents. Thus, the implementation of a project-based approach to patent research and development authorization, customized for individual research endeavors, aligns with market demands and can be operationalized efficiently. This view aligns with the operating efficiency advocated by focus group experts. Such an organization could potentially be managed by academic institutions or industry associations, akin to a Federation of Industry.

### 4.2.3. Sustainable Technological Development through the Project-Based Authorization Business Strategy Framework

The synthesis of the proposed business strategy framework integrates patent analysis, patent pool systems, and project-based operational mechanisms through patent information platforms and authorization systems. Patent authorizations can be adapted to two fundamental formats based on recipient needs: individual authorizations for single patents and package-based authorizations tailored to the developmental requisites of research projects. This business strategy framework encapsulates the concepts of patent commercialization value, as elucidated by Hsueh and Jheng [16], alongside the effects of patent portfolio management highlighted by Hoskisson and Yiu [5] and the principle of complementary patents presented by Sun and Xu [12], as well as strategic entrepreneurship by Xin et al. [23]. Translating the framework into operational mechanisms, the patent supply chain of product series encompassing solar PV systems, silicon materials, silicon wafer materials, and solar PV power converters can be effectively integrated, as depicted in Figure 4.

Figure 4 indicates that effective business strategies can be implemented in response to global competition, especially when the number of patents in the solar cell industry in any country or region reaches an economic scale. These strategies are designed to promote the advancement of industrial research and provide incentives for further research and development in the industry. The operational strategy of a patent pool helps to create an environment conducive to the advancement of solar cell technology within small-to-medium enterprises (SMEs), as well as at an individual level, significantly enhancing the international competitiveness of the solar cell industry. Therefore, the fundamental logic program of this study involves two main steps. The first step is a comprehensive analysis of global solar cell patents. The second step entails establishing a patent information platform and authorization center. The primary focus of this center is to adopt a project-based licensing business strategy, facilitating the effective operation of the patent pool system.

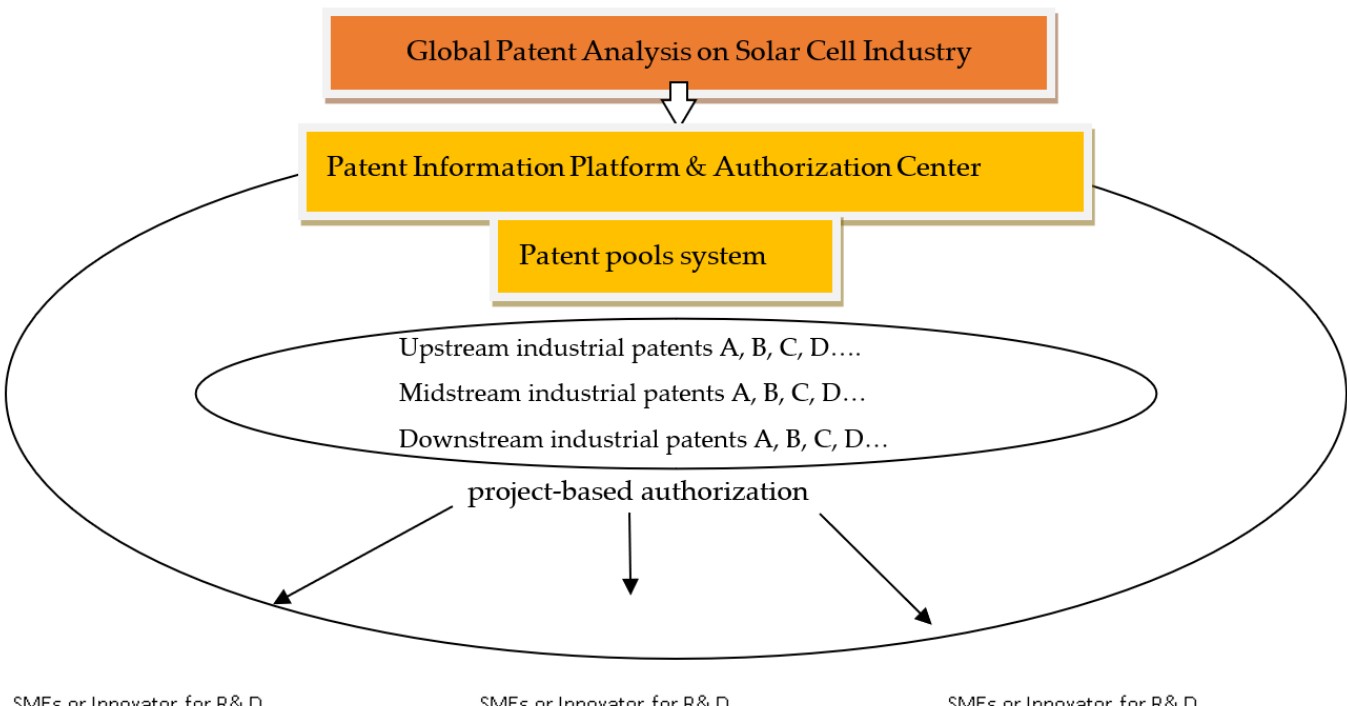

**Figure 4.** A conceptual framework of a patent-pooling business strategy for a solar cell supply chain.

## 5. Conclusions

This study is grounded in expectancy theory and aims to enhance the commercial value of patents for solar cell enterprises and their counterparts within the supply chain. The primary goal is to establish a comprehensive business strategy framework for small and medium-sized enterprises (SMEs) and aspiring innovators operating in the market. The study is driven by the context of limited resources and funding for innovative research and development, to effectively address these challenges. To achieve this, the study conducted an in-depth analysis of the solar cell industry, examining patents across its upstream, midstream, and downstream sectors. Additionally, a thorough patent analysis was conducted to gain insights into the current state of patent development within the solar cell industry.

The subsequent focus of the study was on creating a patent-pooling information platform that complies with US antitrust regulations. We meticulously selected the top 15 countries or regions characterized by a significant market economic scale and a substantial volume of solar cell patents. These patents have been earmarked as the targets for the patent-pooling information platform. In regions or countries where many patents coexist with a robust economic scale, the potential for establishing a new business model that reinvigorates patent utilization is tangible, ultimately enhancing the role of patents in contributing to industrial development. Following a comprehensive analysis and summarization of the patents, we propose the adoption of a project-based licensing approach to cater to the requirements of niche markets and SMEs enterprises. This approach aims to optimize the market value and benefits associated with these patents. The outcomes of the study cater to the needs and expectations of SMEs and enthusiastic innovators aiming to optimize their R&D expenditures. The resulting business strategy framework is designed to support R&D processes for SMEs within the supply chain and for individual innovators, offering streamlined access to comprehensive project packages and detailed piece-by-piece authorization mechanisms. This approach encourages innovation by building upon the achievements of previous research and development, potentially facilitating successful innovations and underscoring the practicality of the research findings.

In terms of real-world application, the study's findings lay the groundwork for establishing and operating patent information platforms and authorization centers across a range

of industries. Through the incorporation of a patent pool system and the implementation of a project-based authorization mechanism, a commercially viable business model can be actualized, contributing to substantial commercial applicability.

This study has several limitations. The research does not delve into the specifics of negotiation with patentees to ensure equitable authorization within the operational context of the proposed framework. Similarly, the practical operational mechanics of the information platform are not explored in detail. Furthermore, the study does not investigate how the benefits derived from reduced transaction costs within the patent pooling are reinvested into the broader community. Interested parties could pursue further research to address these concerns.

**Author Contributions:** Conceptualization, S.-M.L., J.-S.C., C.-L.H. and C.-C.C.; methodology, S.-M.L., J.-S.C., C.-L.H. and C.-C.C.; software, C.-C.C.; validation, S.-M.L., J.-S.C., C.-L.H. and C.-C.C.; formal analysis, S.-M.L. and C.-C.C.; data curation, S.-M.L., J.-S.C. and C.-C.C.; writing—original draft preparation, S.-M.L. and C.-C.C.; writing—review and editing, J.-S.C. and C.-C.C. All authors have read and agreed to the published version of the manuscript.

**Funding:** This research received no external funding.

**Institutional Review Board Statement:** Not applicable.

**Informed Consent Statement:** Informed consent was obtained from all subjects involved in the study.

**Data Availability Statement:** The data provided in this study were generated through the analysis of specific WIPS databases and can be made available upon request from the corresponding author. However, due to intellectual property reasons, this data is not publicly accessible.

**Acknowledgments:** Thanks to Lunghwa University of Science and Technology for providing the WIPS Global patent database.

**Conflicts of Interest:** The authors declare no conflict of interest.

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
