# Peer review of "A Business Strategy Framework for Patent Pooling in the Solar Cell Supply Chain"

_sustainability, doi:10.3390/su152115573_

Round 1
Reviewer 1 Report
Comments and Suggestions for Authors
The manuscript is clear, relevant to the field, and presented in a well-structured manner.
The cited references are mostly recent publications and relevant to research.
There is no evidence of self-citation.
The design of the manuscript is appropriate to test the hypothesis.
The manuscript’s results are reproducible based on the details given in the methods section.
The figures/tables/images/schemes are appropriate
The figures/tables/images/schemes properly show the data.
They are easy to interpret and understand.
The data are interpreted appropriately.
The statistical analysis and data acquired from patent databases are interpreted appropriately.
The conclusions are consistent with the evidence and arguments presented?
Comments on the Quality of English Language
English need minor revision.
Author Response
Thank you for the reviewer's comment. The manuscript will be revised to incorporate the suggested changes and updated results. Subsequently, it will be submitted to the journal's "Language Editing Services".
It has been revised and edited by the journal's English editing service. (23, Oct.)

Reviewer 2 Report
Comments and Suggestions for Authors
First, I believe this is a timely study on eco-friendly energy such as low-carbon emissions such as new and renewable energy, and I offer the following opinions.
1. The introduction summarizes previous studies, but it is not clear from what existing limitations this study began and what fundamental alternatives it provides. Therefore, please clearly present the main purpose by comparing it with existing research.
2. The literature review in Chapter 2 was categorized and organized into three topics, but there is a need to organize the three topics comprehensively to ensure consistency with this study. Additionally, the differentiation of this study should be highlighted.
3. This study uses qualitative analysis as a research method, but this has the limitation that arbitrary structuring is possible without the researcher's objective control. Therefore, a rationale and explanation for the objectification process for qualitative analysis must be provided.
4. Information on basic statistics is required.
5. From the analysis results in Chapter 4, additional explanation is needed on what the source of keyword matching is and how to understand the context of individual keywords.
6.Figure 3 is ultimately seen as the final result of this study, but a detailed explanation of the structure is needed.
7. Please correct the inscriptions and grammatical errors in the sentences.
Comments on the Quality of English LanguageThere is a need to correct sentence inscriptions and grammatical errors.
Author Response
- Add some references to the introduction section, literature review, and analysis section to support the whole manuscript, such as [7], [14], and [21].
- Add a reference, [21], to support this study.
- This research mainly focuses on the quantitative analysis and application of the database system, and then links it to the application of enterprise-side patent analysis and business layout. It is not a purely qualitative research. In order to achieve this important purpose, focus groups were held and industry experts were invited to discuss the feasibility of the project's implementation in the industry and the needs for practical application of business practices from a qualitative perspective. Therefore, the research takes into account both qualitative and quantitative research requirements.
- This study is based on database analysis, which has been explained in Methodology. In order to avoid duplication, the presentation of statistics is added in table 4.
- revised. We do restate the keywords clearly.
- revised. We add the detailed explanation on p.13.
7. The manuscript will be revised to incorporate the suggested changes and updated results. Subsequently, it will be submitted to the journal's "Language Editing Services."

Reviewer 3 Report
Comments and Suggestions for Authors
1. The paper contains redundant words. Please remove unnecessary words.
2. In Figure 2, the symbols and axis labels are too small to read. Could you increase the font size?
3. I couldn't grasp the concept of the Activity Index (AI) from the image. Could you provide the equation and a detailed explanation?
4. How crucial is Patent Pooling for the Business Strategy Framework? Could you provide a reaction figure illustrating the mechanism and explain how Patent Pooling impacts the business market?
Author Response
- The manuscript will be revised to incorporate the suggested changes and updated results. Subsequently, it will be submitted to the journal's "Language Editing Services."
- Corrected, figures reprocessed. (p.10)
3.corrected, see figure 2.(p.10)
The manuscript has been revised and edited by the journal's English editing service. (October 23)
- corrected, Add literature citations, descriptions and Figure 1.(p.4-5)

Reviewer 4 Report
Comments and Suggestions for Authors
1. Line 285: why are there exactly 15 countries listed? It is recommended to explain this number in the manuscript.
2. The text of the manuscript from chapter "2. Literature review" is not aligned
3. The chapter on consultation should be more widely described
4. The manuscript should refer to more literature positions
Author Response
- revised, add description.(p.9)
- thanks.
3.revised. Description added.(p.6)
4.revised. literature added.(p.4-5)
The manuscript has been revised and edited by the journal's English editing service. (October 23)

Reviewer 5 Report
Comments and Suggestions for Authors
I suggest the authors to perform major revisions before resubmitting the article. The second part of the article (Literature Review) does not contain an appropriate literature analysis, as only a few pieces were analysed (while some do not represent the newest findings). Moreover, there needs to be more connection between literature analysis and the empirical part of the research. The conclusions from the empirical part of the research are very general and do not show any relevant findings worth publishing. The conclusion part should be re-done, proposing profound results, scientific discussion, and ideas for further research.
Author Response
- Add some references to the introduction section, literature review, and analysis section to support the discussion, such as [7], [14], and [21].
- Add literature analysis to the description of results.
- Add an explanation of Figure 4.
4. Partial corrections to the conclusion part.

Round 2
Reviewer 2 Report
Comments and Suggestions for Authors
Appropriate changes have been made to the review opinion.
Comments on the Quality of English LanguageSome corrections to terminology and academic expressions are needed.
Author Response
The manuscript has been revised and edited by the journal's English editing service. (October 23)

Reviewer 5 Report
Comments and Suggestions for Authors
n/a
Comments on the Quality of English Languagen/a
Author Response

(The authors gave the same response as above.)
